# GENERATED GRAPH DETECTION

## ABSTRACT

Graph generative models become increasingly effective for data distribution approximation and data augmentation. Although still in sandboxes, they have aroused public concerns about their malicious misuses or misinformation broadcasts, just as what *Deepfake* visual and auditory media has been delivering to society. It is never too early to regulate the prevalence of generated graphs. As a preventive response, we pioneer to formulate the generated graph detection problem to distinguish generated graphs from real ones. We propose the first framework to systematically investigate a set of sophisticated models and their performance in four classification scenarios. Each scenario switches between seen and unseen datasets/generators during testing to get closer to real world settings and progressively challenge the classifiers. Extensive experiments evidence that all the models are qualified for generated graph detection, with specific models having advantages in specific scenarios. Resulting from the validated generality and oblivion of the classifiers to unseen datasets/generators, we draw a safe conclusion that our solution can sustain for a decent while to curb generated graph misuses.

## 1 INTRODUCTION

Graph generative models aim to learn the distributions of real graphs and generate synthetic ones Xie et al. (2022); Liu et al. (2021); Wu et al. (2021b). Generated graphs have found applications in numerous domains, such as social networks Qiu et al. (2018), e-commerce Li et al. (2020), chemoinformatics Kearnes et al. (2016), etc. In particular, with the development of deep learning, graph generative models have witnessed significant advancement in the past 5 years Stoyanovich et al. (2020); Liao et al. (2019); Kipf & Welling (2016); You et al. (2018a).

However, a coin has two sides, there is a concern that the synthetic graphs can be misused. For example, molecular graphs are used to design new drugs Simonovsky & Komodakis (2018); You et al. (2018a). The generated graphs can be misused in this process and it is important for the pharmaceutical factory to vet the authenticity of the molecular graphs. Also, synthetic graphs make deep graph learning models more vulnerable against well-designed attacks. Existing graph-level backdoor attacks Xi et al. (2021) and membership inference attacks Wu et al. (2021a) require the attackers to train their local models using the same or similar distribution data as those for the target models. Adversarial graph generation enables attackers to generate graphs that are close to the real graphs. It facilitates the attackers to build better attack models locally hence keeping those attacks more stealthy (since the attackers can minimize the interaction with the target models). This advantage also applies to the latest graph attacks such as the property inference attack Zhang et al. (2022) and GNN model stealing attack Shen et al. (2022).

As a result, it is essential to regulate the prevalence of generated graphs. In this paper, we propose to proactively target the generated graph detection problem, i.e., to study whether generated graphs can be differentiated from real graphs with machine learning classifiers.

To detect generated graphs, we train graph neural network (GNN)-based classifiers and show their effectiveness in encoding and classifying graphs Zhang et al. (2020); Kipf & Welling (2017); Hamilton et al. (2017). Figure 2 illustrates the general pipeline of the generated graph detection. To evaluate their accuracy and generalizability, we test graphs from varying datasets and/or varying generators that are progressively extended towards the unseen during training. The *seen* concept in dataset or generator means that the graphs used in the training and testing stage are from the same dataset or generated by the same generator, respectively. That is to say, they share the same or similar distribution. And the *unseen* concept represents the opposite.

To sophisticate our solution space, we study three representative classification models. The first model is a direct application of GNN-based end-to-end classifiers Kipf & Welling (2017); Hamilton et al. (2017); Chen et al. (2018); Xu et al. (2019b). The second model shares the spirit of contrastive learning for images Chen et al. (2020); Wu et al. (2018); Hénaff (2020) and graphs Zhu et al. (2021a); You et al. (2020); Hassani & Ahmadi (2020); Zhu et al. (2021b), which, as one of the cutting-edge self-supervised representation learning models, learns similar representations for the same data under different augmentations. We adapt graph contrastive learning to learn similar representations of graphs from the same source under different augmentations. The third model is based on deep metric learning Xing et al. (2002); Schroff et al. (2015); Song et al. (2016), which learns close/distant representations for the data from the same/different classes. We adopt metric learning to learn close/distant representations for graphs from the same/different sources.

We systematically conduct experiments under different settings for all the classification models to demonstrate the effectiveness of our framework. Moreover, we conduct the dataset-oblivious study which mixes various datasets in order to evaluate the influence along the dataset dimension. The evidenced dataset-oblivious property makes them independent of a specific dataset and practical in real-world situations.

## 2 PRELIMINARIES

**Notations.** We define an undirected and unweighted homogeneous graph as $\mathcal{G} = (\mathcal{V}, \mathcal{E}, \mathbf{A})$, where $\mathcal{V} = \{v_1, v_2, ..., v_n\}$ represents the set of nodes, $\mathcal{E} \subseteq \{(v, u) \mid v, u \in \mathcal{V}\}$ is the set of edges and $\mathbf{A} \in \{0, 1\}^{n \times n}$ denotes $\mathcal{G}$'s adjacency matrix. We denote the embedding of a node $u \in \mathcal{V}$ as $\mathbf{h}_u$ and the embedding of the whole graph $\mathcal{G}$ as $\mathbf{h}_\mathcal{G}$.

**Graph Neural Networks.** Graph neural networks (GNNs) have shown great effectiveness in fusing information from both the graph topology and node features Zhang et al. (2020); Hamilton et al. (2017); Kipf & Welling (2017). In recent years, they become the start-of-the-art technique serving as essential building blocks in graph generators and graph classification algorithms. A GNN normally takes the graph structure as the input for message passing, during which the neighborhood information of each node $u$ is aggregated to get a more comprehensive representation $\mathbf{h}_u$. The detailed information of GNN is described in Appendix A.1.

**Graph Generators.** Graph generators aim to produce graph-structured data of observed graphs regardless of the domains, which is fundamental in graph generative models. The study of graph generators dated back at least to the work by Erdös-Rényi Erdös & Rényi (1959) in the 1960s. These traditional graph generators focus on various random models Erdös & Rényi (1959); Albert & Barabási (2002), which typically use simple stochastic generation methods such as a random or preferential attachment mechanism. However, the traditional models require prior knowledge to obtain/tune the parameters and tie them specifically to certain properties (e.g., probability of connecting to other nodes, etc.), hence their limited capacity of handling the complex dependencies of properties. Recently, graph generators using GNN as the foundation have attracted huge attention Liao et al. (2019); You et al. (2018b); Grover et al. (2019); Simonovsky & Komodakis (2018). The GNN-based graph generators can be further grouped into two categories: autoencoder-based generators and autoregressive-based generators. Autoencoder-based generator Kipf & Welling (2016); Grover et al. (2019); Mehta et al. (2019); Simonovsky & Komodakis (2018) is a type of neural network which is used to learn the representations of unlabeled data and reconstruct the input graphs based on the representations. Autoregressive-based generator Liao et al. (2019); You et al. (2018b) uses sophisticated models to better capture the properties of observed graphs. By generating graphs sequentially, the models can leverage the complex dependencies between generated graphs. In this paper, we selectively focus on eight graph generators that span the space of commonly used architectures, including ER Erdös & Rényi (1959), BA Albert & Barabási (2002), GRAN Liao et al. (2019), VGAE Kipf & Welling (2016), Graphite Grover et al. (2019), GraphRNN You et al. (2018b), SBMGNN Mehta et al. (2019), and GraphVAE Simonovsky & Komodakis (2018) (see more detailed information about graph generators in Appendix A.2).

# 3 GENERATED GRAPH DETECTION

## 3.1 PROBLEM STATEMENT

The generated graph detection problem studied in this paper can be formulated as follows. Suppose we have a set of *real* graphs $\mathbf{RG} = \{\mathbf{rg}_1, \ldots, \mathbf{rg}_\ell\}$, $m$ *seen* graph generators $\Phi_{seen} = \{\phi_1, \ldots, \phi_m\}$, $k$ *unseen* graph generators $\Phi_{unseen} = \{\phi_{m+1}, \ldots, \phi_{m+k}\}$, and a collection of *generated graphs* by seen and unseen generators $\mathbf{GG} = \{\mathbf{GG}_1, \ldots, \mathbf{GG}_{m+k}\}$. Here each $\mathbf{GG}_i$ is a set of graphs generated by a graph generator $\phi_i$. To be specific, let $D = \{(x_1, y_1), (x_2, y_2), \ldots, (x_z, y_z)\}$, where $x_i \in \mathbf{RG} \bigcup \mathbf{GG}$, $y_i$ represents the label of each graph (i.e., real or generated) and $z = \sum_{i=1}^{m+k} |\mathbf{GG}_i|$ is the total number of samples. A generated graph detector $f(\cdot)$ is later trained on $D$. Once trained, it classifies each testing graph as real or generated. However, it is normal to see the arrival of graphs from unknown generators that have never been seen in training, and the graphs may not bear with similar properties as of the training data in the real world. The existing solutions usually leverage model retraining to cope with the problem. Yet, it is impractical to retrain a model from scratch every time a new graph generator is added or unseen data is encountered. Ideally, $f(\cdot)$ *should be built in a way that it can be generalized to previously unseen data/generator in the real world*.

## 3.2 A GENERAL FRAMEWORK FOR GENERATED GRAPH DETECTION AND ANALYSIS

In this paper, we propose a general framework to detect generated graphs. Specifically, this framework consists of four scenarios depending on whether the dataset or graph generator has been used to train the model. These scenarios comprehensively cover from the simplest close world scenario to the most challenging full open world detection scenario. We discuss how we choose different ML models to implement this framework in Section 3.3.

**Closed World.** In this scenario, the training and testing graphs are sampled from *seen datasets* and generated by *seen generators*. The goal is to predict whether a graph is real or generated by seen generators. Under this setting, to train the generated graph detector $f(\cdot)$, we sample real graphs from $\mathbf{RG}$ as positive samples and sample graphs generated by seen generators $\Phi_{seen}$ as negative samples. The graphs used to test $f(\cdot)$ share the same distribution with the training set, i.e., they consist of real graphs sampled from $\mathbf{RG}$ and generated graphs generated by seen generators $\Phi_{seen}$.

**Open Generator.** In this scenario, the negative samples of the testing graphs are generated by unseen generators but are in the same or similar distribution of training data (i.e., seen data). The training data of $f(\cdot)$ does not contain any graphs generated by these unseen generators. Since only the graph generators used in the testing dataset are not seen at the time of training, we thus name it "Open Generator" scenario. Under this setting, the detector $f(\cdot)$ is trained with the graphs sampled from $\mathbf{RG}$ (positive samples) and graphs generated by seen generators $\Phi_{seen}$ (negative samples). The positive samples used to test $f(\cdot)$ are also from $\mathbf{RG}$ while the negative samples are generated by unseen generators $\Phi_{unseen}$. The goal is to predict whether a graph is real or generated by unseen generators.

**Open Set.** In this scenario, the testing graphs are from seen generators that are trained on an unseen dataset. Concretely, the graph generators that the system sees in training are what it will see in testing (i.e., seen generators). However, the testing graphs are of the different distribution of training data (i.e., unseen data). For instance, $f(\cdot)$ was trained using both real and generated graphs from chemical graphs, yet, the testing graphs (either real or generated) are social network graphs that are inherently different. As such, we name this scenario "Open Set" scenario. Similar to the "Open Generator" scenario, the detector $f(\cdot)$ is trained with the graphs sampled from $\mathbf{RG}$ (as positive samples) and the graphs generated by seen generators $\Phi_{seen}$ (as negative samples). Unlike previous experiment, the graphs used to test $f(\cdot)$ are from different datasets. The testing graphs of $f(\cdot)$ consist of real graphs sampled from other datasets and graphs generated by seen generators $\Phi_{seen}$ based on other datasets. The goal is to predict whether a graph from unseen dataset is real or generated.

**Open World.** In this scenario, the testing graphs are from unseen generators that are trained on an unseen dataset. This setting is the most challenging yet common in the real world. It is normal to see the arrival of graphs from unknown generators that have never been seen at the training time, and the graphs may not bear with similar properties as of the training data. To be specific, certain generators that the system sees at the testing time are not included in its training stage (i.e., unseen generators), and the testing graphs are of the different distribution of training data (i.e., unseen data).

Similar to "Open Generator" and "Open Set" scenarios, the generated graph detector $f(\cdot)$ is trained with the graphs sampled from $\mathbf{RG}$ (as positive samples) and the graphs generated by seen generators $\Phi_{seen}$ (as negative samples). The testing graphs consist of real graphs sampled from other datasets and graphs generated by unseen generators $\Phi_{unseen}$ based on other datasets. The goal is to predict whether a graph from unseen dataset is real or generated by unseen generators.

## 3.3 DETECTION METHODOLOGIES

As discussed in Section 3.2, we need an ML model $f(\cdot)$ to cope with the four generated graph detection scenarios. In this section, we introduce three ML models – end-to-end classifier Kipf & Welling (2017); Hamilton et al. (2017); Chen et al. (2018); Xu et al. (2019b), contrastive learning-based model Zhu et al. (2021a); You et al. (2020); Hassani & Ahmadi (2020); Zhu et al. (2021b), and metric learning-based model Xing et al. (2002); Schroff et al. (2015); Song et al. (2016) – to implement the aforementioned detection framework. All the models can work as the $f(\cdot)$ to do the final detection in all scenarios. For each scenario, $f(\cdot)$ has the same structure, while trained or tested by different samples. The pros and cons of each model are evaluated and discussed in Section 4.

**End-to-end Classifier.** The most straightforward approach to distinguishing between real and generated graphs is to train a binary classifier in an end-to-end manner. As aforementioned, among all the research, graph classification methods based on graph convolutional networks (GCNs) are commonly recognized as the state-of-the-art technique in deep learning-based graph classification Kipf & Welling (2017); Hamilton et al. (2017); Chen et al. (2018); Xu et al. (2019b). Also, we can also see the results from Appendix A.5 also shows that GCN performs better in most of the times compared with other GNN networks. Therefore we choose the GCN model Kipf & Welling (2017) as our end-to-end classifier. It consists of four GCN layers and a fully connected layer. We use this four-layer GCN network to embed the graph data into a 128-dimensional vector, and use a fully connected layer to compute the final classification result.

**Contrastive Learning-based Model.** Previous studies have shown that contrastive learning helps to improve the graph encoding performance Zhu et al. (2021a); You et al. (2020); Hassani & Ahmadi (2020); Zhu et al. (2021b). Different from the traditional binary classifier that trains the GNN model in an end-to-end manner, the contrastive learning-based model first learns a powerful graph encoder in a self-supervision manner, then uses the graph encoder to transform the graphs into graph embeddings, and employs a binary classifier to predict the results. Figure 3 (in Appendix A.3) illustrates the general workflow of our contrastive learning-based model. We use support vector machines (SVM) as the final classifier following the previous work Sun et al. (2020a); You et al. (2020). The implementation details of the contrastive learning-based model are introduced in Appendix A.3.

**Metric Learning-based Model.** In the past few years, deep metric learning has consistently achieved the state-of-the-art model performance Xing et al. (2002); Schroff et al. (2015); Song et al. (2016). As one of the cutting-edge unsupervised representation learning models, deep metric learning aims to map input data into a metric space, where data from the same class get close while data from different classes fall apart from each other. However, unlike other tasks such as classification or face recognition in which only one training sample is needed to get the output, in metric learning, at least two training samples are needed at one time, as the output of metric learning is whether the two input samples are from the same category Guo et al. (2017); He et al. (2018). Based on the core concept of metric learning, siamese network Guo et al. (2017); He et al. (2018) is proposed, which takes paired samples as inputs and outputs whether the paired samples are from the same category. The implementation details of metric learning-based model is introduced in Appendix A.4. Since the siamese network takes paired samples as input and only predicts whether the two input samples are from the same label, to evaluate the performance of the metric learning-based model in the perspective of getting prediction results of each graph, we still need to predict the exact label for each testing sample by querying the siamese network using paired samples consists of one testing sample and one known sample.

In order to get the final classification results, for each testing sample, we randomly select $N_k$ samples of each label from the training set and generate $N_k * N_{class}$ paired samples. Here $N_{class}$ equals to 2 (i.e., real and generated). After feeding the paired samples into the siamese network, we will get $N_k$ posteriors for each label. Each posterior represents the probability of the paired samples from the same label. After calculating the mean value of the $N_k$ posteriors for each label, we can find the

maximum mean value and take the corresponding label as the predicted result of the testing sample. For example, if the maximum mean value of the posteriors is from label *real*, then we consider using *real* as the final classification result.

## 4 EXPERIMENTS

We first introduce the datasets and implementation details of our experiments. Then following the application scenarios described in Section 3, we conduct experiments based on each scenario.

### 4.1 EXPERIMENTAL SETUP

**Datasets.** We use 7 benchmark datasets from TUDataset Morris et al. (2020) to evaluate the performance, including AIDS Riesen & Bunke (2008), Alchemy Chen et al. (2019), Deezer ego nets (abbreviated as Deezer) Rozemberczki et al. (2020), DBLP DBL, GitHub StarGazer (abbreviated as GitHub) Rozemberczki et al. (2020), COLLAB Yanardag & Vishwanathan (2015) and Twitch ego nets (abbreviated as Twitch) Rozemberczki et al. (2020).

Among them, Deezer, GitHub, and Twitch are social networks with nodes representing users and edges indicating friendships. DBLP and COLLAB are collaboration networks with nodes representing papers/researchers and edges indicating citations/collaborations. AIDS and Alchemy are molecular graphs with nodes representing atoms of the compound and edges corresponding to chemical bonds. These graphs form our *real* datasets for the rest of the evaluation. The statistics of all the datasets are summarized in Table 1.

Table 1: Dataset statistics.

| Dataset | # of graphs | Avg. Nodes | Avg. Edges |
|---------|------------|------------|------------|
| AIDS | 2,000 | 15.69 | 16.20 |
| Alchemy | 202,579 | 10.10 | 10.44 |
| Deezer | 9,629 | 23.49 | 65.25 |
| DBLP | 19,456 | 10.48 | 19.65 |
| GitHub | 12,725 | 113.79 | 234.64 |
| COLLAB | 5,000 | 74.49 | 2,457.78 |
| Twitch | 127,094 | 29.67 | 86.59 |

**Sampling High-quality Generated Graphs.** Although the graph generators are capable to generate graphs with similar distribution as real graphs, some of the generated graphs may still contain obvious artifacts in some cases. There is a concern that the classification may be biased by such artifacts. Thus we compute the number of nodes, the number of edges, density, diameter, average clustering and transitivity as the statistical features of each graph and use Euclidean Distance to measure the 1-nearest-neighbor similarity between each generated graphs and real graph sets Yu et al. (2019). We select 20% generated graphs with the highest similarity for the following experiments.

To evaluate the quality of generated graphs, we use maximum mean discrepancy (MMD) over these graph features to measure the similarity between real graphs and graphs generated by different generators. The MMD results show that the graphs generated by different generators and real graphs are statistically indistinguishable. The MMD results are shown in Table 6 in Appendix A.6.

**Implementation Details.** We use the GCN to embed the graphs in the end-to-end classifier and metric learning-based model. The GCN is implemented in PyTorch PyT. The optimizer we used is Adam optimizer Kingma & Ba (2015). Each model is trained for 200 epochs. The learning rate is set to 0.001 and we adopt Cross-Entropy Loss as the loss function. The ratio of the training set and testing set is 8:2. The contrastive learning-based model is trained following the implementation details in GraphCL You et al. (2020). As mentioned in Section 3.3, we generate $N_{ps} * 2$ paired samples to train the siamese network in the metric learning-based model and use $N_k$ samples from each label to predict the final results. $N_{ps} * 2$ is the number of paired samples used to train the siamese network. Here we conduct experiments to fine-tune metric learning-based model and find the best $N_{ps} = 200,000$ and $N_k = 10$ which makes the model perform the best. The corresponding results are displayed in Appendix A.6 (Figure 5 and Figure 6).

**Baseline.** To better evaluate the performance of our proposed models, We incorporate a new model named Feature Classification (FC) as the baseline. FC model leverages the graph statistical features as input and uses Multilayer perceptron (MLP) to do the final prediction. The statistical features we used are the number of nodes, the number of edges, density, diameter, average clustering and transitivity, which are the same as the features we used to sample high-quality graphs.

## 4.2 Experiments for the "Closed World" Scenario

In this scenario, we want to explore whether real graphs and generated graphs can be distinguished when the distribution of all testing graphs are known. As introduced before, we propose three methods to classify graphs. The evaluation metrics we used in this paper are accuracy and F1-score.

**Overall Results.** The accuracy of binary classification is summarized in Table 3. In general, our proposed models outperform FC model in all datasets, demonstrating that the GNN-based models can better capture the characteristics of graphs compared to using MLP with only statistical features. Also, we observe that among the three methods, the metric learning-based model performs the best in most cases, while the contrastive learning-based model performs least satisfying. Moreover, the results show that in general, the performance of Deezer, Github, and Twitch is better than other datasets. Compared to other datasets, the graphs in Twitch, Github, and Deezer are bigger and the three datasets also have a richer amount of graphs. This implies that the binary classifiers can distinguish between real graphs and generated graphs with higher accuracy for larger datasets with bigger graphs.

Although the contrastive learning-based model and metric learning-based model have a similar goal, i.e. training an encoder that makes graphs with the same label get closer to each other and graphs with different labels fall apart, the metric learning-based model performs better than the contrastive learning-based model in this scenario. Thus we can draw the conclusion that the embeddings produced by the metric learning tend to be distinguished easily in graph datasets.

**Dataset Oblivious Study.** Besides the evaluation from the perspective of a single dataset, we also conduct the dataset oblivious study. In this experiment, we first randomly sample 1,000 real graphs from each dataset. Then we randomly select 1,000 generated graphs which are evenly generated by all the generators from each dataset. Finally, we obtain a *mixed* dataset consisting of 7,000 real graphs and 7,000 generated graphs to train the binary classifier.

Surprisingly, a persuasive performance can also be noticed even when we don't take the dimension of the dataset into consideration. The performance indicates that the models can still distinguish real graphs from generated graphs even when the graphs used to train the model don't belong to any specific dataset. This is more meaningful in the real world scenario as we may not know which dataset the graphs come from at the test time. In the mixed dataset, the end-to-end classifier performs the best, which means when the graphs which need to be classified do not belong to one specific dataset, the end-to-end classifier can better capture the complex dependencies of graphs and detect generated graphs with higher accuracy.

**Distinguish Graphs Generated by Unseen Generators.** Apart from classifying real graphs and graphs generated by unseen generators, we use metric learning to predict whether two graphs generated by unseen generators are generated by the same generator. To evaluate the performance of predicting whether any two graphs generated by unseen generators are generated by the same generator, we randomly generate 50,000 positive graph pairs and 50,000 negative graph pairs and use metric learning to take the graph pairs as input (the performance is shown in Table 2). It can be noticed that the metric learning can predict whether two graphs generated by unseen generators are generated by the same generator to some extent. Moreover, we can see that the performance of Deezer, Github and TWITCH are better than other datasets, which is consistent with the results of the "Open Generator" scenario.

Table 2: Distinguish Graphs Generated by Unseen Generators.

|  | Accuracy | F1-score |
|---|---|---|
| AIDS | 0.78 | 0.78 |
| Alchemy | 0.82 | 0.82 |
| Deezer | 0.93 | 0.93 |
| DBLP | 0.75 | 0.74 |
| GitHub | 0.95 | 0.95 |
| COLLAB | 0.83 | 0.83 |
| TWITCH | 0.89 | 0.89 |
| MIXED | 0.64 | 0.64 |

However, when we use the mixed dataset to train and test the metric learning-based model, the performance is much worse than in other datasets. It is reasonable since we can see from Table 3 that the metric learning-based model with a mixed dataset performs the worst among all the datasets. The visualization of graphs generated by the unseen generators shown in Figure 7 also supports our results, the embeddings of the mixed dataset can not be separated explicitly compared to other datasets.

Table 3: The accuracy/F1-score of generated graph detection in "Closed World" scenario and "Open Generator" scenario.

| Dataset | Closed World | | | | Open Generator | | | |
|---|---|---|---|---|---|---|---|---|
| | FC | End-to-end | Contrastive | Metric | FC | End-to-end | Contrastive | Metric |
| AIDS | 0.75/0.73 | 0.89/0.85 | 0.87/0.84 | **0.91/0.90** | 0.73/0.70 | 0.82/0.81 | 0.84/0.82 | **0.87/0.84** |
| Alchemy | 0.78/0.78 | 0.87/0.87 | 0.85/0.80 | **0.90/0.89** | 0.74/0.73 | 0.80/0.77 | 0.82/0.79 | **0.84/0.82** |
| Deezer | 0.78/0.78 | 0.97/0.95 | 0.95/0.94 | **0.98/0.97** | 0.74/0.74 | 0.90/0.88 | **0.92/0.92** | 0.91/0.91 |
| DBLP | 0.70/0.68 | **0.84/0.83** | 0.82/0.82 | 0.82/0.82 | 0.75/0.74 | 0.79/0.79 | **0.82/0.82** | 0.80/0.79 |
| Github | 0.81/0.81 | 0.95/0.94 | 0.92/0.92 | **0.96/0.96** | 0.80/0.82 | 0.94/0.94 | 0.91/0.91 | **0.96/0.92** |
| COLLAB | 0.56/0.55 | 0.85/0.84 | 0.84/0.82 | **0.89/0.89** | 0.50/0.49 | 0.78/0.76 | 0.80/0.79 | **0.84/0.82** |
| Twitch | 0.56/0.55 | 0.92/0.89 | 0.90/0.88 | **0.95/0.93** | 0.51/0.49 | 0.85/0.85 | **0.90/0.89** | 0.86/0.86 |
| Mixed | 0.64/0.62 | **0.84/0.83** | 0.80/0.80 | 0.82/0.81 | 0.60/0.59 | 0.78/0.76 | **0.82/0.80** | 0.79/0.78 |

### 4.3 EXPERIMENTS FOR THE "OPEN GENERATOR" SCENARIO

The experiments above have proved that the real graphs and graphs generated by different generators can be distinguished in the close world scenario. In order to further evaluate if our models can still detect generated graphs when given unseen generators, we choose three different generators - GraphRNN You et al. (2018b), SBMGNN Mehta et al. (2019), and GraphVAE Simonovsky & Komodakis (2018) - to generate fake graphs for each dataset. For all datasets, we use real graphs as the positive samples and graphs generated by unseen generators as the negative samples to test the models.

**Overall Results.** The final classification results are shown in Table 3, from which we can see that the binary classification results of all the datasets are over 0.75, which indicates that even when the graphs are generated by unseen algorithms, the models can still have a relatively good performance. This indicates that all the models can generalize to other generators. Also, we can see a higher accuracy of the metric learning-based model, which exemplifies that it can better generalize to other generators.

**Dataset Oblivious Study.** Moreover, we also train all models for the mixed dataset. It can be noticed that the performance of the mixed dataset is in part with those of other datasets. The experimental results suggest that our models can still generalize to previously unseen generators even when we don't take the dimension of the dataset into consideration.

The accuracy of the contrastive learning-based model for the mixed dataset is even better for COLLAB and is the best among the three models. This suggests that the contrastive learning-based model can better generalize to other generators in datasets with a wide range of node numbers and graph densities, i.e. mixed dataset.

### 4.4 EXPERIMENTS FOR THE "OPEN SET" SCENARIO

Apart from distinguishing between real graphs and graphs generated by unseen algorithms, we also conduct experiments to evaluate whether graphs generated by unseen datasets can still be distinguished from real graphs. In this experiment, we use graphs from AIDS, Alchemy, Deezer, DBLP, and Github as the seen datasets to train all the models, and use COLLAB and Twitch as unseen datasets to test the models.

For each seen dataset, we randomly select 1,000 real graphs and 1,000 generated graphs which are evenly generated by the seen generators. In the end, we use the final dataset with 5,000 real graphs and 5,000 generated graphs to train all the models.

In this scenario, we want to evaluate whether the fake graphs generated by seen generators in unseen datasets can be distinguished from the real graphs. To test the model, for each unseen dataset, we randomly select real graphs and the same amount of generated graphs which are evenly generated by the seen generators. The final testing set contains 2,000 real graphs and 2,000 generated graphs. The performance of all the models is summarized in Table 4.

We can see from the table that, in general, the real graphs and generated graphs can be distinguished with an accuracy higher than 0.78. This implies that our models have the ability to generalize to unseen datasets. Moreover, the accuracy of the contrastive learning-based model is higher than 0.85

Table 4: Generated graph detection in "Open Set" scenario and "Open World" scenario.

| Metric | Open Set | | Open World | |
|---|---|---|---|---|
| | Accuracy | F1-score | Accuracy | F1-score |
| Feature classification | 0.57 | 0.54 | 0.64 | 0.62 |
| End to end classifier | 0.82 | 0.82 | 0.76 | 0.75 |
| Contrastive learning-based model | **0.85** | 0.84 | **0.83** | 0.83 |
| Metric learning-based model | 0.78 | 0.76 | 0.74 | 0.74 |

and the best among the three models, which suggests that the contrastive learning-based model can generalize to the unseen datasets better.

After comparing the performance with the "Closed World" scenario in Section 4.2, we find that the performance drops. It is reasonable because the graphs used to test the models come from new datasets which are not seen in the training set, which makes the task harder than in the previous experiment.

### 4.5 Experiments for the "Open World" Scenario

The fourth scenario is to evaluate whether the fake graphs generated by unseen algorithms in unseen datasets can be distinguished from the real graphs. To test the model, for each unseen dataset, we randomly select real graphs and the same amount of generated graphs which are evenly generated by the unseen generators. The final testing set contains 2,000 real graphs and 2,000 generated graphs. The performance of all the models is summarized in Table 4.

The scenario is called the "Open World" scenario as described before since the datasets and generators are both unseen in the training phase. It is the hardest task among the four scenarios. We can see from Table 4 that the performance, as expected, is lower than those in Section 4.4 and Section 4.5.

Although the performance is not as competent, the accuracies of all models are still higher than 0.74. This suggests that the models can still distinguish real graphs and graphs generated by unseen generators in unseen datasets to some extent. Apart from that, the contrastive learning-based model performs the best among all the models, which is in line with the previous experiments.

Throughout all the experiments, we can draw a conclusion that the metric learning-based model tends to perform better in the "Closed World" scenario while the contrastive learning-based model shows advantages in "Open Generator", "Open Set" and "Open World" scenarios. The results give us an insight that metric learning can learn better representations of graphs with known graph distributions. On the contrary, as a representative self-supervised method, the contrastive learning-based model can learn representations that are more general and can be transferred to different graph distributions.

### 4.6 Visualization Analysis

Among the previous experiment, we can draw the conclusion that contrastive learning-based models tend to perform better in "Open Generator", "Open Set" and "Open World" scenarios with the mixed dataset, we further explore the reason behind it. To this end, we use t-Distributed Stochastic Neighbor Embedding (t-SNE) van der Maaten & Hinton (2008) to visualize the graphs embedded by different models. Figure 1 shows the t-SNE results of the testing samples used in the fourth scenario. It can be easily noticed that the embeddings produced by the contrastive encoder can be divided better, which may be the major reason why the contrastive learning-based model outperforms other models in the "Open World" Scenario.

## 5 Related work

We have already covered several highly-related works (e.g., graph generative models and graph neural networks) in Section 2. We discuss additional related work in a broader scope below.

**Generated Data Detection.** Although it remains an unexplored area in generated graph detection, there has been some research about generated image detection in the past few years. Rössler et al.

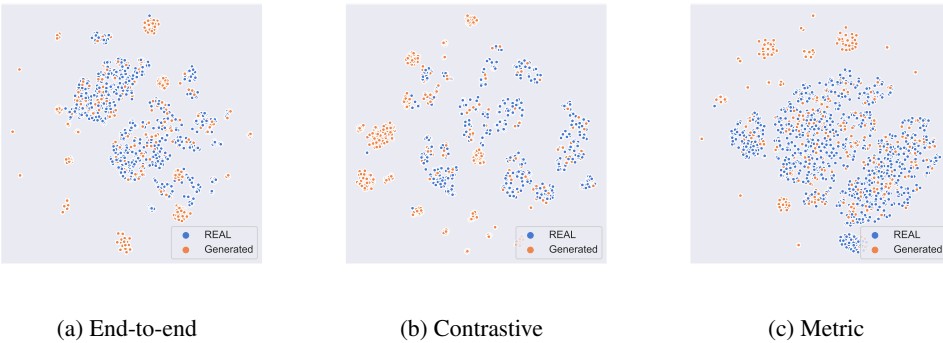

| (a) End-to-end | (b) Contrastive | (c) Metric |

Figure 1: The visualization results of different models in "Open World" scenario.

showed that simple classifiers can detect images created by a single category of networks Rössler et al. (2019). Wang et al. Wang et al. (2020) demonstrated that a simple image classifier trained on one specific CNN generator (ProGANKarras et al. (2018)) is able to generalize well to unseen architectures. Ning et al. Yu et al. (2019) learned the GAN fingerprints towards image attribution and showed that even a small difference in GAN training (e.g., the difference in initialization) can leave a distinct fingerprint that can be detected. Most of the previous studies focus on image data; as far as we know, we are the first to investigate the generated graph detection.

**Privacy and Security Issues in GNN.** Rising concerns about the privacy and security of GNNs have led to a surge of research on graph adversarial attacks. Broadly speaking, they can be grouped into two categories — causative attacks and exploratory attacks. Causative attacks on GNNs add unnoticeable adversarial perturbations to node features and graph structures to reduce the accuracy of or intentionally change the outcome of node classification Bojchevski & Günnemann (2019); Caverlee et al. (2020); Ma et al. (2020); Sun et al. (2020b); Wu et al. (2019); Xu et al. (2019a), link prediction Bojchevski & Günnemann (2019); Lin et al. (2020), graph classification Dai et al. (2018); Xi et al. (2021), etc. To conduct causative attacks, attackers must be able to tamper with the training process of GNNs or influence the fine-tuning process of pre-trained GNNs. Exploratory attacks on GNNs send (carefully crafted) query data to the target GNNs and observe their decisions on these input data. Attackers then leverage the responses to build shadow models to achieve different attack goals, such as link re-identification He et al. (2021), property inference Zhang et al. (2022), membership inference Wu et al. (2020), model stealing Duddu et al. (2020), etc. To launch exploratory attacks, attackers must be able to interact with the GNNs (e.g., via publicly accessible API) at the runtime.

## 6 CONCLUSION AND FUTURE WORK

In this paper, we propose a general framework for generated graph detection. In this framework, we introduce four application scenarios based on different training data and testing data and design three kinds of models to accomplish experiments in each scenario. The experimental results show that all models can distinguish real graphs from generated graphs successfully in all scenarios, which means that although the generative models show great advantage and success in many domains, the generated graphs can still be detected by GNN-based models. Also, we notice that the metric learning-based model tends to perform the best in the close world scenario while the contrastive learning-based model always shows the best performance in "Open Generator", "Open Set" and "Open World" scenarios, which suggests that the contrastive learning-based model can generalize to datasets and generators better. Our experiment about dataset oblivious study shows that our models can still work with a persuasive performance when we use the mixed dataset to train and test the models. This is an interesting finding since the graphs in different datasets vary a lot, hence the mixed dataset tends to have a wide range of node numbers and densities. The results imply that our models can handle datasets with many disparate graphs. The finding also fits more to the real world situation, where the graphs that need to be detected may not be from a specific dataset. Moreover, although we only discuss the detection of generated graphs in this paper, the framework can also be extended to other research areas, such as images, text, or audio.

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

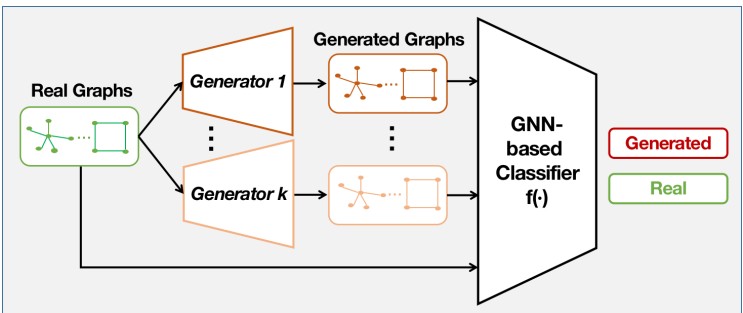

Figure 2: **The pipeline of generated graph detection.** The real graphs are from real world datasets, the generated graphs are generated by different graph generators based on real graphs. The GNN-based classifier is built to classify real graphs and generated graphs.

# A APPENDIX

## A.1 THE DETAILED INFORMATION OF GRAPH NEURAL NETWORKS

Graph Neural Networks (GNNs) have shown great effectiveness in fusing information from both the graph topology and node features Zhang et al. (2020); Hamilton et al. (2017); Kipf & Welling (2017). In recent years, they become the start-of-the-art technique as essential building blocks in graph generators and graph classification algorithms.

**General Definition.** Most of the GNNs learn the node representations for graph-structured data by a neighborhood aggregation strategy, where a model iteratively updates the representation of a node through message passing and aggregating representations of its neighbors. After $k$ iterations of aggregation, we can get the node's representation $h_v$ which stores the structural information within its $k$-hop neighborhood. Typically, the GNN contains multiple graph convolutional layers. The definition of each layer is as follows:

$$\mathbf{h}_v^k = \phi\left(\mathbf{h}_v^{k-1}, \psi\left(\mathbf{h}_v^{k-1}, \mathbf{h}_u^{k-1}\right)\right), \forall u \in \mathcal{N}(v), \tag{1}$$

where $\mathcal{N}(v)$ is a set of nodes adjacent to node $v$. $\mathbf{h}_v^{k-1}$ is the node embedding of node $u$ after $k$ iterations, $\psi\left(\mathbf{h}_v^{k-1}, \mathbf{h}_u^{k-1}\right)$ represents the message received from the neighbours, and $\phi(\cdot)$ is an aggregation operation.

**Aggregator.** Recently, researchers have proposed different kinds of practical implementations of aggregation operations Kipf & Welling (2017); Hamilton et al. (2017); Chen et al. (2018); Xu et al. (2019b), among which the Graph Convolutional Networks (GCN) is the most representative method which uses the symmetric normalization method to aggregate all the information from the neighbors and shows great success Kipf & Welling (2017). The aggregation process of GCN can be defined as follows:

$$\mathbf{h}_v^k = \phi\left(\mathbf{h}_u^{k-1}, u \in \mathcal{N}(v) \cup v\right) = \sum_{u \in \mathcal{N}(v) \cup v} \mathbf{h}_j^{(k-1)} / \sqrt{d_u d_v} \tag{2}$$

where $d_u$ and $d_v$ are the node degrees of node $u$ and $v$, respectively. Here $\phi(\cdot)$ is a mean aggregation operator.

**Graph Pooling.** After obtaining the embeddings of all nodes, we use a graph pooling operation to integrate the embeddings of all nodes in the graph to get the embedding of the whole graph. In our graph classification model, we use a straightforward but efficient approach called *mean pooling* that averages all the node embeddings to obtain the graph embedding, i.e., $\mathbf{h}_\mathcal{G} = \frac{1}{|\mathcal{V}|} \sum_{u \in \mathcal{V}} \mathbf{h}_u$.

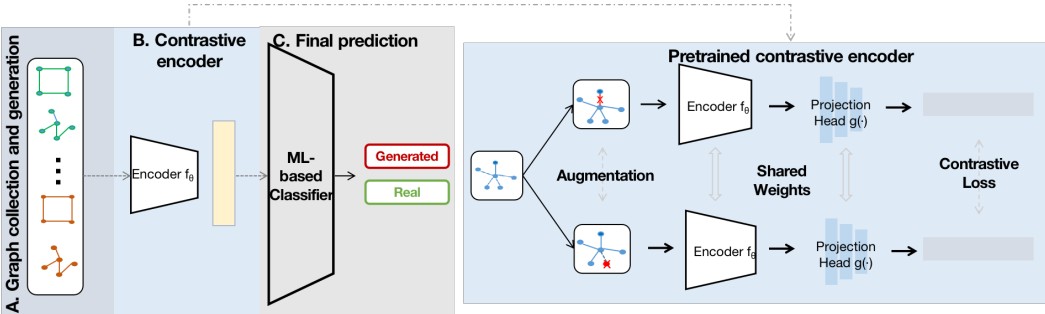

Figure 3: **The workflow of the contrastive learning-based model.** The model trains a contrastive encoder to embed the graphs and uses a machine learning-based classifier to do the final classification. The contrastive encoder is based on GraphCL You et al. (2020), which uses the graph augmentation method to get two correlated augmented views as a positive pair and embed them with a GNN-based encoder $f(\cdot)$ and projection head $g(\cdot)$. Then the contrastive loss function is used to maximize the agreement between the positive pairs. After embedding all the graphs, we use support vector machines (SVM) to do the final prediction.

## A.2 THE DETAILED INFORMATION OF GRAPH GENERATORS

**Traditional Graph Generator.** Erdös-Rényi (ER) model Erdös & Rényi (1959) and Barabási-Albert (BA) model Albert & Barabási (2002) are two commonly used traditional graph generators. Given a few parameters, these generators can be explicitly expressed by formulas. ER model generates random graphs with a fixed number of nodes and edges. BA model is often used to generate scale-free graphs using a preferential attachment mechanism. That is, a new node will be added each time, and the edges will be randomly added to connect the new node and the existing nodes.

**Autoencoder-based Generator.** The autoencoder-based generator is a type of neural network which is used to learn the representations of unlabeled data and reconstruct the input graphs based on the representations. We consider VGAE Kipf & Welling (2016), Graphite Grover et al. (2019), SBMGNN Mehta et al. (2019), and GraphVAE Simonovsky & Komodakis (2018) in this paper. VGAE uses a graph convolutional network (GCN) as the encoder and the inner product as the decoder. The model can obtain the node features and capture the overall distribution of input graphs. Based on VGAE, Grover et al. proposed Graphite, a latent variable generative model which also utilizes GCN as the encoder. Unlike VGAE, Graphite adds a multi-layer iterative neural network before the inner product to construct the decoder. SBMGNN produces graphs by modeling sparse latent variables, which makes it competitive in preserving the community structure. It uses a sparse variational autoencoder (VAE) Kingma & Welling (2014) to model graphs. The decoder consists of a fast recognition model which models the probability of an edge between two nodes by a nonlinear function. GraphVAE is also an autoencoder based-model. It uses a feed-forward network with edge-conditioned graph convolutions (ECC) Simonovsky & Komodakis (2017) to encode the graphs into continuous representations. The main idea of the decoder is to output the probabilistic fully-connected graph and at last use a standard graph matching algorithm to align it to the original graph.

**Autoregressive-based Generator.** To capture the complex dependencies of all nodes and edges, autoregressive-based generators are proposed. An autoregressive-based generator adds nodes and edges sequentially. In this paper, we include GRAN Liao et al. (2019) and GraphRNN You et al. (2018b) as the Autoregressive-based model. GRAN generates a block of nodes and associated edges at each step. It uses GNN with attention to utilizing the topology of the generated part of the graph, which makes GRAN model the dependencies between the already generated part and the newly generated part more effectively. GraphRNN uses two recurrent neural networks (RNN), which are called *graph-level* RNN and *edge-level* RNN. The *graph-level* RNN is used to maintain the state of a generated graph and generate new nodes. The *edge-level* RNN is used to generate the edges between new nodes and the already existing graph.

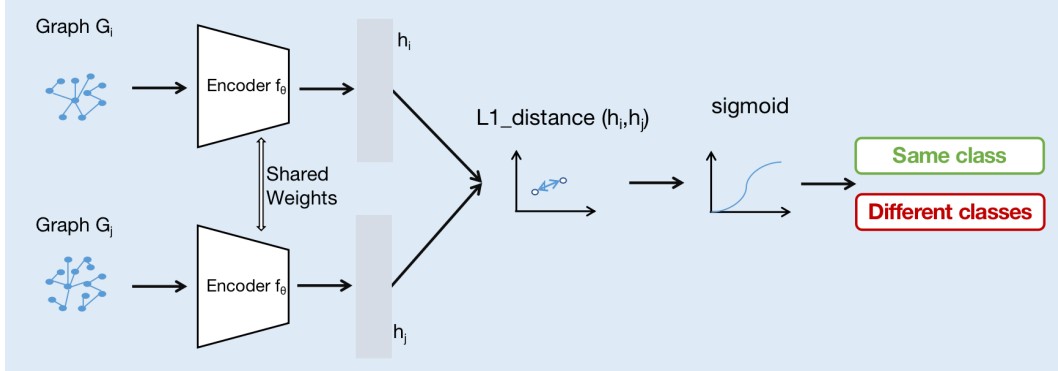

Figure 4: The workflow of the siamese network. In siamese network, two graphs $G_i$ and $G_j$ are fed into the encoder $f_\theta$ one by one to get the embeddings $h_i$ and $h_j$. The L1 distance between $h_i$ and $h_j$ are calculated and used to do the final prediction.

## A.3 THE IMPLEMENTATION DETAILS OF CONTRASTIVE LEARNING-BASED MODEL

**Training Contrastive Encoder.** As one of the cutting-edge unsupervised representation learning models, contrastive learning aims at learning similar representations of the same data under different augmentations. It is widely used in visual representation learning Chen et al. (2020); Wu et al. (2018); Hénaff (2020) and graph embedding Zhu et al. (2021a); You et al. (2020); Hassani & Ahmadi (2020); Zhu et al. (2021b). In this paper, we use GraphCL You et al. (2020) as the graph contrastive encoder.[1]

In the graph contrastive encoder, we first use the graph augmentation method to get two correlated augmented views $\hat{\mathcal{G}}_i$ and $\hat{\mathcal{G}}_j$ as a positive pair. We use node dropping as the first augmentation, and randomly select one augmentation from the augmentation pool as the second one. The augmentation pool consists of node dropping, edge perturbation, and subgraph. Then the $\hat{\mathcal{G}}_i$ and $\hat{\mathcal{G}}_j$ will be embedded by a GNN-based encoder $f_\theta$. After that, a projection head $g(\cdot)$ is used to map the embeddings to a different latent space to calculate the contrastive loss. The projection head $g(\cdot)$ used in contrastive learning is a multi-layer perceptron (MLP). Then the contrastive loss function is used to maximize the agreement between the positive pairs. Here the loss function is the normalized temperature-scaled cross-entropy loss (NT-Xent) Sohn (2016); Wu et al. (2018).

When training the contrastive encoder, a mini-batch of $N$ graphs will be randomly sampled and processed, which means that $2 * N$ augmented graphs and the corresponding loss need to be optimized each time. We denote the augmented $n$th graph in the mini-batch as $z_{n,i}$, $z_{n,j}$ and use them as the positive pairs. The negative pairs are generated by $z_{n,i}$, $z_{n,j}$ and other augmented graphs within the same mini-batch except for $z_{n,i}$, $z_{n,j}$. Finally, the NT-Xent for the $n$th graph is defined as follows:

$$\text{sim}\left(\boldsymbol{z}_{n,i}, \boldsymbol{z}_{n,j}\right) = \frac{\boldsymbol{z}_{n,i}^\top \boldsymbol{z}_{n,j}}{\|\boldsymbol{z}_{n,i}\| \|\boldsymbol{z}_{n,j}\|} \tag{3}$$

$$\text{Loss} = -\log \frac{\exp\left(\text{sim}\left(\boldsymbol{z}_{n,i}, \boldsymbol{z}_{n,j}\right)/\tau\right)}{\sum_{n'=1, n'\neq n}^{N} \exp\left(\text{sim}\left(\boldsymbol{z}_{n,i}, \boldsymbol{z}_{n',j}\right)/\tau\right)} \tag{4}$$

where $\text{sim}\left(\boldsymbol{z}_{n,i}, \boldsymbol{z}_{n,j}\right)$ means the cosine similarity of $z_{n,i}$, $z_{n,j}$. $\tau$ denotes the temperature parameter. After generating the loss for each graph in the mini-batch, the overall loss is computed across all the positive pairs in the mini-batch.

## A.4 THE IMPLEMENTATION DETAILS OF METRIC LEARNING-BASED MODEL

**Training Siamese Network.** Figure 4 shows the workflow of the siamese network. Two graphs $G_i$ and $G_j$ are fed into the encoder $f_\theta$ one by one to get the embeddings $h_i$ and $h_j$. In the siamese

---

[1]The source code of GraphCL is in `https://github.com/Shen-Lab/GraphCL`.

Table 5: The accuracy of different backbones in end-to-end model.

| Dataset | GAT | GIN | GCN |
|---------|-----|-----|-----|
| AIDS | 0.89 | 0.88 | 0.89 |
| Alchemy | 0.88 | 0.86 | 0.87 |
| Deezer | 0.96 | 0.96 | 0.97 |
| DBLP | 0.86 | 0.82 | 0.84 |
| Github | 0.93 | 0.94 | 0.95 |
| COLLAB | 0.84 | 0.82 | 0.85 |
| Twitch | 0.91 | 0.91 | 0.92 |
| Mixed | 0.82 | 0.83 | 0.84 |

network, the paired samples are fed into the same encoder and the weights of them are shared. Then the L1 distance $d$ between the two embeddings is calculated by the following equation.

$$d = abs(||h_i - h_j||) \tag{5}$$

$d$ is used to feed into the loss function and tune the network to get a better embedding. The loss function we used in this paper is called binary cross-entropy loss, which is commonly used in classification tasks. The equation of binary cross equation loss is as follows:

$$Loss = -(y \log(p) + (1 - y) \log(1 - p)) \tag{6}$$

$d$ is used to feed into the loss function and tune the network to get a better embedding. The loss function we used in this paper is called binary cross-entropy loss, which is commonly used in classification tasks.

To train the siamese network, we sample a training set that consists of $Num_{ps}$ paired samples with the same label and $Num_{ps}$ paired samples with different labels for each dataset.

## A.5   ABLATION STUDY

**Different GNN Networks.** Our end-to-end model uses GCN as the backbone. We replace it with GIN Xu et al. (2019b) and GAT Velickovic et al. (2018), and compare the performance of different backbones. We can see from Table 5 that in most of the time, GCN performs better than the others, thus we choose GCN as the final backbone of end-to-end model in our experiment.

**Metric Learning-based Model..** To achieve the best classification results of the metric learning-based model, we evaluate the impact of different $N_{ps}$ and $N_k$ on model performance. Here $N_{ps}$ represents the number of paired samples used to train the siamese network. $N_k$ denotes the reference samples used to obtain the final prediction results.

We can see from Figure 5 that if we use more paired samples to train the siamese network, the metric learning-based model tends to perform better. Due to the time and resource limitations, we finally choose to use 20,000 paired samples to train the siamese network. Figure 6 shows that in general, the metric learning-based model shows the best performance when $N_k = 10$. Thus we use $N_k = 10$ in the following experiments.

## A.6   ADDITIONAL RESULTS

In the appendix, plots are illustrated for additional information and experimental results as mentioned throughout the paper.

**MMD results.** Table 6 shows the MMD results of real graphs and graphs generated by different generators.

**The visualization results of graphs generated by unseen generators..** Figure 7 shows the visualization results of three unseen generators (GraphRNN, GRAPHVAE and SBMGNN) produced by the metric learning-based model.

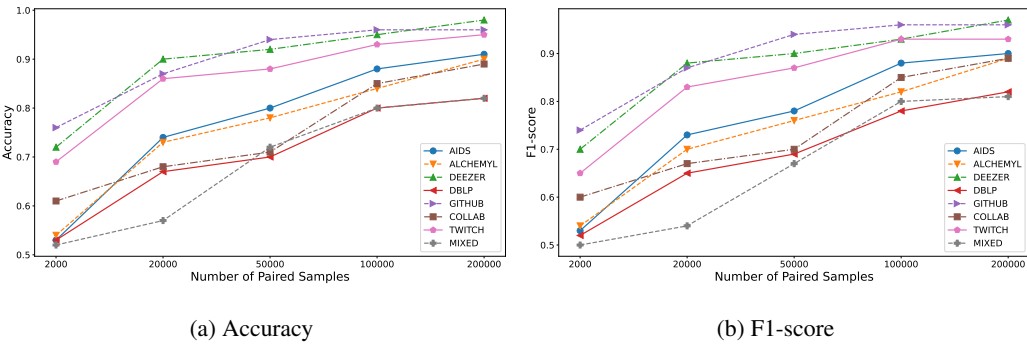

(a) Accuracy

(b) F1-score

Figure 5: **Different** $N_{ps}$. The impact of different number of paired samples on metric learning-based model.

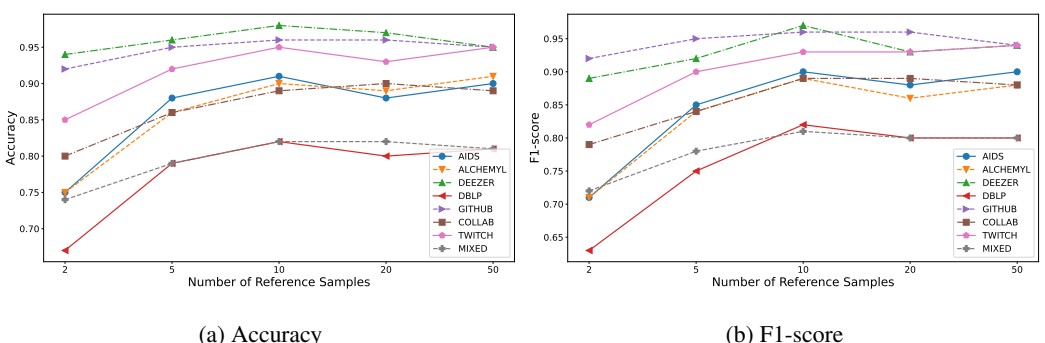

(a) Accuracy

(b) F1-score

Figure 6: **Different** $N_k$. The impact of different number of reference samples on metric learning-based model.

Table 6: The MMD results of real graphs and graphs generated by different generators.

|  | ER | BA | graphite | VGAE | GRAN | GraghRNN | GraghVAE | sbmgnn |
|---|---|---|---|---|---|---|---|---|
| AIDS | 0.0863 | 0.1070 | 0.0547 | 0.0324 | 0.0534 | 0.1016 | 0.0526 | 0.1884 |
| Alchemy | 0.0139 | 0.1167 | 0.0205 | 0.0088 | 0.0641 | 0.1493 | 0.0284 | 0.2844 |
| Deezer | 0.0356 | 0.1161 | 0.1869 | 0.2565 | 0.1168 | 0.0755 | 0.1167 | 0.1363 |
| DBLP | 0.0601 | 0.3117 | 0.0759 | 0.1168 | 0.4496 | 0.3835 | 0.1808 | 0.0238 |
| Github | 0.0249 | 0.2513 | 0.0398 | 0.0452 | 0.0315 | 0.1635 | 0.0314 | 0.0294 |
| COLLAB | 0.0270 | 0.0362 | 0.0092 | 0.0118 | 0.0024 | 0.0099 | 0.0021 | 0.0175 |
| Twitch | 0.0148 | 0.0336 | 0.0728 | 0.0589 | 0.0182 | 0.0663 | 0.0218 | 0.0681 |

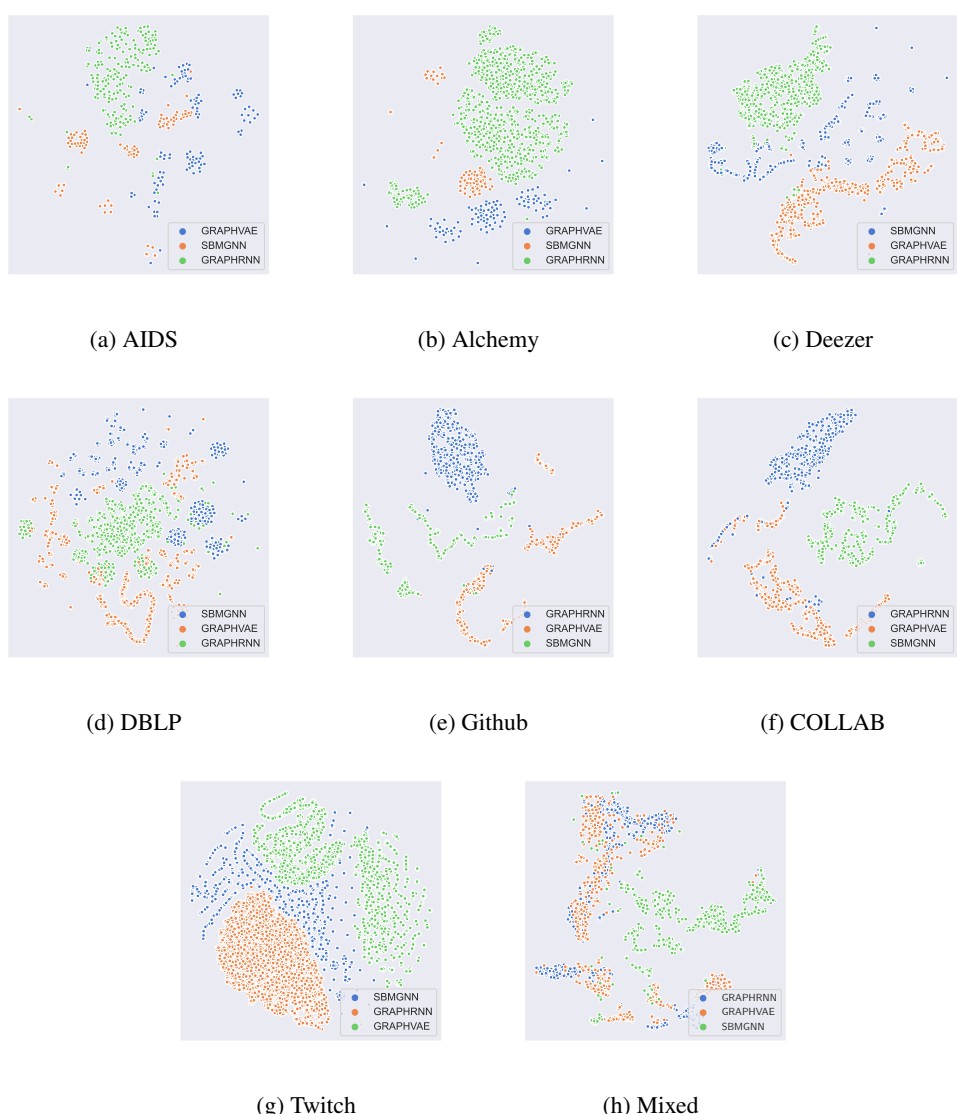

(a) AIDS        (b) Alchemy        (c) Deezer

(d) DBLP        (e) Github        (f) COLLAB

(g) Twitch        (h) Mixed

Figure 7: The visualization results of graphs generated by unseen generators.

