# OpenReview forum: "Generated Graph Detection"
_ICLR.cc/2023/Conference — Submitted to ICLR 2023_

### Official Review · Reviewer_eg3n · 2022-10-23

**Confidence:** 4
**Correctness:** 4
**Technical Novelty And Significance:** 3
**Empirical Novelty And Significance:** 3
**Recommendation:** 6

**Clarity, Quality, Novelty And Reproducibility:**

The paper is clear and novel. It is also straightforward to replicate the experiments.

**Details Of Ethics Concerns:**

As I said in my review: it is possible to use this to actually devise adversarial attacks to GNN algorithms. This mighr represent a possible issue for society.

**Strength And Weaknesses:**

# Strengths

1. Novel problem
2. Method is simple enough to be easy to implement


# Weaknesses

1. The method itself could be used to detect generated graphs that cannot be detected easily and thus being used in an attack
2. The classifiers that  authors tested on their datasets are pretty much "standard". I wonder wether there might be some better architecture that can be used to take specific care of this task

**Summary Of The Paper:**

The paper discusses the problem of detecting graphs that have been generated algorithmically. The paper discusses four different types of scenarios from the simplest (where the model knows all the possible details about the generation process) to the hardest (where nothing is known in advance).

The classifier used is an end-to-end trained GNN encoder with an MLP used to output the classification outcome.

The classifier has a decent performance, although not super high making the task challenging.

**Summary Of The Review:**

Some questions to authors:

1. When you write: "... use Euclidean Distance to measure the 1-nearest-neighbor similarity between each generated graphs and real graph sets..." ==> how can you make sure that euclidean in the embedding space is the right distance? Have a look at these two papers (https://www.sciencedirect.com/science/article/abs/pii/S002002551630696X ; https://arxiv.org/abs/1812.09095) is there anything that you could use to better compute the 1-nearest neightbor?

2. If you have a good way to detect generated graphs, wouldn't this be useful to filter out generated graphs that are difficult to spot? How would you prevent this from happening?

---

> ### Author Response · Authors · 2022-11-09
> **Response to Reviewer eg3n:**
>
> We thank all the reviewers for their careful and constructive comments, which help us improve our work and make our paper more solid.
> And we are happy to receive positive feedback/comments on the following aspects:
>
> - Novel problem (Reviewer chCC, eg3n)
> - Experiments on both ideal circumstances and realistic circumstances (Reviewer chCC, 4WuZ)
> - Adequate evaluation (Reviewer 4WuZ)
> - Provided some practical insights (Reviewer 4WuZ)
> - Easy to replicate the experiments (Reviewer eg3n)
> - The writing is clear (Reviewer chCC, 4WuZ, eg3n)
>
> The concerns of Reviewer 4WuZ are addressed in the following.
>
> 1. Using different distance measures.
>
> - We will try the two different graph distance measures and add the experimental results later.
>
> 2. If you have a good way to detect generated graphs, wouldn't this be useful to filter out generated graphs that are difficult to spot? How would you prevent this from happening?
>
> - The main purpose of this paper is to find the optimal method to discern real and generated graphs under different scenarios. Our results indicate that although the distance between real and generated graphs is relatively small (which means that they share similar distribution), the existing GNN-based classifiers can still distinguish between real and generated graphs. That is to say, it is our target to find a way to filter out (classify) generated graphs that are difficult to spot to prevent the misuse of generated graphs. Could you please be more specific about why should we prevent this from happening?

---

### Official Review · Reviewer_4WuZ · 2022-10-25

**Confidence:** 5
**Correctness:** 3
**Technical Novelty And Significance:** 1
**Empirical Novelty And Significance:** 2
**Recommendation:** 3

**Clarity, Quality, Novelty And Reproducibility:**

Originality: The proposed methods are not novel. The main novelty in the paper is proposing the problem of discerning between real graphs and fake graphs. I argue this is not a big intellectual/technical leap from the analogous problem in image/video data.

Clarity: The paper is very clear. The notation, proposed problem, datasets, and evaluation are all clear to me.

Quality: I believe the evaluation is of above average scientific quality. The technical quality is limited, as none of the methods are novel.

**Strength And Weaknesses:**

Strengths:
- Experimental results provide some practical insights on the performance of different ML approaches for the target task
- Authors explore 4 scenarios of increasing complexity, which provides a valuable comparison on how the models perform under ideal circumstances vs. more realistic circumstances.

Weaknesses:
- Unclear significance and relevance of the addressed problem. The authors' argument is that they are preemptively studying a problem that may become relevant in the future. It is challenging to assess significance at this moment. It is also challenging to assess whether the performance obtained by the models would be good or not in practice for this hypothetical task. Having a case study would help justify the authors' claims of significance.
- No novelty. The evaluation and data generation is performed using existing methods.

**Summary Of The Paper:**

DISCLAIMER: I reviewed this paper for NeurIPS 2022 earlier this year. As far as I can tell, not much (if anything) has changed in this manuscript. As such, I will reiterate the points in my original review.

Summary:

The authors investigate the performance of existing graph neural network models to discern between real graphs and "fake" graphs generated synthetically, potentially for a nefarious purpose. The authors explore 4 scenarios of increasing complexity and real-world relevance, as well as 3 different approaches from the machine learning literature to learn to discern between real and fake graphs.

**Summary Of The Review:**

See my comments above. My main concern with this paper is the lack of novelty. If the main contribution of the paper is a new ML task, I do not find compelling evidence in the paper to suggest that this task will have an application in the future. It may very well be the case that this task will become relevant for a certain domain at some point. But we simply do not know, and we can just "extrapolate" from other types of data, such as images.

I am curious as to how the authors arrived to the generated graph detection problem? Was there a particular application in mind where fake graphs are indeed a concern? If so, please discuss in the paper, as it would *significantly* strengthen the contribution. Or was the thought more along the lines of "if the task of detecting fake images is of interest to the community, why not detecting fake graphs instead"?

If the idea of a new task is indeed the crux of the contribution, I would recommend this paper to be accepted as a short letter, an extended abstract, or a poster, not as a full research paper.

---

> ### Author Response · Authors · 2022-11-09
> **Response to Reviewer 4WuZ:**
>
> We thank all the reviewers for their careful and constructive comments, which help us improve our work and make our paper more solid.
> And we are happy to receive positive feedback/comments on the following aspects:
>
> - Novel problem (Reviewer chCC, eg3n)
> - Experiments on both ideal circumstances and realistic circumstances (Reviewer chCC, 4WuZ)
> - Adequate evaluation (Reviewer 4WuZ)
> - Provided some practical insights (Reviewer 4WuZ)
> - Easy to replicate the experiments (Reviewer eg3n)
> - The writing is clear (Reviewer chCC, 4WuZ, eg3n)
>
> The concerns of Reviewer 4WuZ are addressed in the following.
>
> 1. Unclear significance and relevance of the addressed problem.
>
> - Similar to adversarial sample detection in other domains, the main concern about adversarial samples is that they can be extended to miscreant actions. We believe that the generated graphs can be misused in the following two aspects. (1) In the chemistry area, the design of new drugs often relies on large-scale molecular graphs of existing drugs[1][2]. The generated graphs can be misused in this process and it is important for the pharmaceutical factory to vet the authenticity of the molecular graphs. (2) On the other side, synthetic graphs make deep graph learning models more vulnerable to well-designed attacks. For example, in adversarial attacks, the synthetic graphs can be used to train and thus make the model misclassify the test point into a specific target class. Large-scale synthetic graphs which share similar distribution with real graphs can make the GNN model more vulnerable to such attacks[3][4][5].
>
> 2. No novelty. The evaluation and data generation is performed using existing methods.
>
> - It is correct that we leverage existing methods to identify fake graphs. However, it does not diminish our novelty in other aspects.
> 	- The research question is novel and important. To our best knowledge, we are the first to address generated graph detection. Thus this paper can be seen as an adoption of adversarial sample detection in graph learning space. It is important to spur further research in the graph learning space.
> 	- Our work is a novel connection that bridges two independently developing research regimes: fake graph generation and fake graph detection. Our work preemptively blocks the potential abuse of graph generative models.
> 	- Our paper proposed a novel experimental protocol/framework to validate the efficacy of generated graph detection. Especially, the open set and open-world scenarios make our work practically usable in the real world.
>
> 3. How the authors arrived to the generated graph detection problem?
>
> - We have seen many successful attacks leveraging adversarial graphs [5][6][7][8][9][10][11]. These attacks include graph-level attacks [12][13][14] and node-level attacks[6][15][16][17]. Therefore, it is urgent to investigate practical countermeasures to mitigate these attacks. One of the promising research directions is to explore the intrinsic difference between adversarial edges/nodes and clean edges/nodes [18][19]. Apparently, it is a game of cat and mouse. The attackers simply make sure that their adversarial graphs are as close to the clean graphs as possible. The graph generation algorithms can be misused exactly for this purpose hence our motivation.
>
> [1] Simonovsky, Martin, and Nikos Komodakis. "Graphvae: Towards generation of small graphs using variational autoencoders." International conference on artificial neural networks. Springer, Cham, 2018.
>
> [2] You, Jiaxuan, et al. "Graphrnn: Generating realistic graphs with deep auto-regressive models." International conference on machine learning. PMLR, 2018.
>
> [3] Wu, Huijun, et al. "Adversarial examples on graph data: Deep insights into attack and defense." arXiv preprint arXiv:1903.01610 (2019).
>
> [4] Zügner, Daniel, Amir Akbarnejad, and Stephan Günnemann. "Adversarial attacks on neural networks for graph data." Proceedings of the 24th ACM SIGKDD international conference on knowledge discovery & data mining. 2018.
>
> [5] Sun, Lichao, et al. "Adversarial attack and defense on graph data: A survey." arXiv preprint arXiv:1812.10528 (2018).
>
> [6] Zügner, Daniel, Amir Akbarnejad, and Stephan Günnemann. "Adversarial attacks on neural networks for graph data." Proceedings of the 24th ACM SIGKDD international conference on knowledge discovery & data mining. 2018.
>
> [7] Zhu, Dingyuan, et al. "Robust graph convolutional networks against adversarial attacks." Proceedings of the 25th ACM SIGKDD international conference on knowledge discovery & data mining. 2019.
>
> [8] Pham, Trang, et al. "Column networks for collective classification." Thirty-first AAAI conference on artificial intelligence. 2017.
>
> [9] Ma, Jiaqi, Shuangrui Ding, and Qiaozhu Mei. "Black-box adversarial attacks on graph neural networks with limited node access." arXiv preprint arXiv:2006.05057 (2020).
>
> [10] Veličković, Petar, et al. "Graph attention networks." arXiv preprint arXiv:1710.10903 (2017).

---

> > ### Author Response · Authors · 2022-11-09
> > **Continued**
> >
> > [11] Xu, Keyulu, et al. "Representation learning on graphs with jumping knowledge networks." International conference on machine learning. PMLR, 2018.
> >
> > [12] Ma, Yao, et al. "Attacking graph convolutional networks via rewiring." arXiv preprint arXiv:1906.03750 (2019).
> >
> > [13] Dai, Hanjun, et al. "Adversarial attack on graph structured data." International conference on machine learning. PMLR, 2018.
> >
> > [14] Xi, Zhaohan, et al. "Graph backdoor." 30th USENIX Security Symposium (USENIX Security 21). 2021.
> >
> > [15] Wu, Huijun, et al. "Adversarial examples on graph data: Deep insights into attack and defense." arXiv preprint arXiv:1903.01610 (2019).
> >
> > [16] Xu, Kaidi, et al. "Topology attack and defense for graph neural networks: An optimization perspective." arXiv preprint arXiv:1906.04214 (2019).
> >
> > [17] Wang, Binghui, and Neil Zhenqiang Gong. "Attacking graph-based classification via manipulating the graph structure." Proceedings of the 2019 ACM SIGSAC Conference on Computer and Communications Security. 2019.
> >
> > [18] Ioannidis, Vassilis N., Dimitris Berberidis, and Georgios B. Giannakis. "Graphsac: Detecting anomalies in large-scale graphs." arXiv preprint arXiv:1910.09589 (2019).
> >
> > [19] Xu, Xiaojun, et al. "Characterizing malicious edges targeting on graph neural networks." (2018).

---

### Official Review · Reviewer_chCC · 2022-10-25

**Confidence:** 3
**Correctness:** 3
**Technical Novelty And Significance:** 2
**Empirical Novelty And Significance:** 2
**Recommendation:** 5

**Clarity, Quality, Novelty And Reproducibility:**

This paper is well-written and well-organized. It is easy to follow the proposed ideas and technological details.
The novelty of this paper is limited.

**Strength And Weaknesses:**

Strengths,

S1: Authors explore four scenarios, which provide both ideal circumstances and realistic circumstances.
S2: This paper proposes a new problem.

Weaknesses,

W1: The motivation is not strong and seems artificial. It is not clear how the generated graphs are used in malicious misuses or misinformation broadcasts.

W2: The novelty is limited. The proposed methods are based on existing works.

**Summary Of The Paper:**

The authors present a new problem, how to detect real graphs and "fake" graphs. The authors then explore four scenarios where the detection algorithm applies, and propose three different machine-learning methods to detect fake graphs.

**Summary Of The Review:**

I recommend a weak reject for this paper because of its limited novelty and the weak motivation.

---

> ### Author Response · Authors · 2022-11-09
> **Response to Reviewer chCC:**
>
> We thank all the reviewers for their careful and constructive comments, which help us improve our work and make our paper more solid.
> And we are happy to receive positive feedback/comments on the following aspects:
>
> - Novel problem (Reviewer chCC, eg3n)
> - Experiments on both ideal circumstances and realistic circumstances (Reviewer chCC, 4WuZ)
> - Adequate evaluation (Reviewer 4WuZ)
> - Provided some practical insights (Reviewer 4WuZ)
> - Easy to replicate the experiments (Reviewer eg3n)
> - The writing is clear (Reviewer chCC, 4WuZ, eg3n)
>
> The concerns of Reviewer chCC are addressed in the following.
>
> 1. The motivation is not strong and seems artificial. It is not clear how the generated graphs are used in malicious misuses or misinformation broadcasts.
>
> - We believe that the generated graphs can be misused in the following two aspects. (1) In the chemistry area, the design of new drugs often relies on large-scale molecular graphs of existing drugs[1][2]. The generated graphs can be misused in this process and it is important for the pharmaceutical factory to vet the authenticity of the molecular graphs. (2) On the other side, synthetic graphs make deep graph learning models more vulnerable to well-designed attacks. For example, in adversarial attacks, the synthetic graphs can be used to train and thus make the model misclassify the test point into a specific target class. Large-scale synthetic graphs which share similar distribution with real graphs can make the GNN model more vulnerable to such attacks[3][4][5].
>
> 2. The novelty is limited. The proposed methods are based on existing works.
>
> - It is correct that we leverage existing methods to identify fake graphs. However, it does not diminish our novelty in other aspects.
> 	- The research question is novel and important. To our best knowledge, we are the first to address generated graph detection. Thus this paper can be seen as an adoption of adversarial sample detection in graph learning space. It is important to spur further research in the graph learning space.
> 	- Our work is a novel connection that bridges two independently developing research regimes: fake graph generation and fake graph detection. Our work preemptively blocks the potential abuse of graph generative models.
> 	- Our paper proposed a novel experimental protocol/framework to validate the efficacy of generated graph detection. Especially, the open set and open-world scenarios make our work practically usable in the real world.
>
>
> [1] Simonovsky, Martin, and Nikos Komodakis. "Graphvae: Towards generation of small graphs using variational autoencoders." International conference on artificial neural networks. Springer, Cham, 2018.
>
> [2] You, Jiaxuan, et al. "Graphrnn: Generating realistic graphs with deep auto-regressive models." International conference on machine learning. PMLR, 2018.
>
> [3] Wu, Huijun, et al. "Adversarial examples on graph data: Deep insights into attack and defense." arXiv preprint arXiv:1903.01610 (2019).
>
> [4] Zügner, Daniel, Amir Akbarnejad, and Stephan Günnemann. "Adversarial attacks on neural networks for graph data." Proceedings of the 24th ACM SIGKDD international conference on knowledge discovery & data mining. 2018.
>
> [5] Sun, Lichao, et al. "Adversarial attack and defense on graph data: A survey." arXiv preprint arXiv:1812.10528 (2018).

---

### Decision · Program_Chairs · 2023-01-20

**Decision:**

Reject

**Justification For Why Not Higher Score:**

This is an interesting idea that needs a more compelling case study and more technical depth.

**Justification For Why Not Lower Score:**

N/A

**Metareview: Summary, Strengths And Weaknesses:**

The work considers the task of learning to detect generated graphs (fake graphs). More specifically, the work investigates the performance of existing graph neural network models to discern between real graphs and generated graphs. The authors explore four scenarios of increasing complexity and real-world relevance, as well as three different approaches from the machine learning literature to learn to discern between real and fake graphs.

The overall impression of the reviewers:
- This is an interesting task
- But an interesting task that needs a clear case study.
- Adversary should adapt to defense
- No novelty in the techniques (the idea of putting these methods together is novel, but reviewers also wanted to see some technical innovation)